# Microfluidics and Organoids, the Power Couple of Developmental Biology and Oncology Studies

**DOI:** 10.3390/ijms241310882

**Published:** 2023-06-29

**Authors:** Laura Ann Hetzel, Ahmed Ali, Vincenzo Corbo, Thomas Hankemeier

**Affiliations:** 1Leiden Academic Center for Drug Research, Leiden University, 2333 CC Leiden, The Netherlands; l.a.hetzel@lacdr.leidenuniv.nl; 2Department of Diagnostics and Public Health, ARC-Net Research Centre, University of Verona, 37134 Verona, Italy; vincenzo.corbo@univr.it

**Keywords:** microfluidic device, organoids, organoid formation, organoid filtering

## Abstract

Organoids are an advanced cell model that hold the key to unlocking a deeper understanding of in vivo cellular processes. This model can be used in understanding organ development, disease progression, and treatment efficacy. As the scientific world embraces the model, it must also establish the best practices for cultivating organoids and utilizing them to the greatest potential in assays. Microfluidic devices are emerging as a solution to overcome the challenges of organoids and adapt assays. Unfortunately, the various applications of organoids often depend on specific features in a device. In this review, we discuss the options and considerations for features and materials depending on the application and development of the organoid.

## 1. Introduction

Advances in disease treatments, developmental biology, and underlying mechanisms associated with each are being limited by the accuracy of the experimental models used. Two dimensional (2D) cell models are the gold standard for research and clinical applications; however, organoids are paving a new path for research. The definition of organoid is continuing to develop over time. An early definition of an organoid is a small tumor cell nest that maintains histological patterns [1]. Today, the definition involves more complexity in the parameters. An organoid, as the name might imply, must resemble an organ, which designates the cell types, required and desired functions, and self-organization processes and patterns that form the three dimensional (3D) structure [2]. The source of cells used to produce the organoid may vary and include an organ slice, a tissue sample, a stem cell, or primary cells [1,2,3,4,5]. It is possible that some 3D structures may lack the complexity to technically be classified as an organoid, however, they are clearly not a 2D culture; these can be referred to as spheroids [5].

Patient-derived organoids (PDOs) are three dimensional (3D) cultures of primary cells that self-organize through cell-to-cell and cell-to-matrix interactions to recapitulate characteristics of the native tissue. PDOs have been shown to preserve critical structural and molecular aspects of the primary tumors, as opposed to traditional monolayer cell culture [6]. Even rare histopathological features of the native tissue are preserved in the derived PDO, which also show extensive genetic conservation by capturing key drivers of the original tumors [6]. PDOs make it possible to appreciate the level of heterogeneity among patients of a specific disease as well as disease subtypes [7]. It is important to note that organoids may be developed from tumor or healthy tissue samples, depending on the purpose of the study, as studying healthy tissue will increase knowledge regarding development and function of organs [8]. Additionally, a drug response study showed that some treatments had responses different between the organoid lines derived from different patients and these responses were consistent with the primary tumor, highlighting intra-tumor heterogeneity in a population and the need for personalized medicine [6]. Similarly, spheroids are being produced from stem cells to mimic human organ development and function; however, caution must be taken as it is unclear if there are limitations in the developmental similarities between organoids and organs [9].

Aside from features of the organoids themselves, the tumor microenvironment will also influence the behavior of the cells, and therefore the response to treatment [10]. Two dimensional models do not initiate the same cell-cell and cell-extra cellular matrix (ECM) interactions that are present in the 3D models [10]. For cell-cell interactions, it is imperative for the cells to interact with various cell types as they would in vivo, however, 2D models do not support this level of complexity while 3D models do [11]. The immune cells that are present in the human body have a profound effect on disease progression and reaction to treatment, and the organoids can be cultured with immune cells present to replicate this effect [11]. As previously mentioned, organoids may be derived to study healthy tissue and diseases other than cancer, so other cocultures, such as those with bacterial microbes, are also critical.

Organoids provide a wealth of advantages; however, they also present significant challenges which can be overcome with microfluidic devices. Microfluidic devices are also used for organs-on-chips, which, similar to organoids, are used to mimic human physiology [12]. The organ-on-chip cells often do not involve a matrix and often do not have to self-organize, creating different demands on the chip [12]. Some of chips discussed may be applicable for use with organ-on-chip; however, organoids and spheroids will be the main focus.

There are two main approaches to creating a microfluidic platform for organoids: the device can be designed for use with single cells to induce formation of the organoids, or it can be designed for fully formed organoids to be injected into the device for filtering or manipulation. There are several established methods for cultivating organoids, which translates into options for formation within a device [13]. Regardless of cultivation taking place on or off the chip, to choose the best methods, factors such as organoid size, isolation, and fluid flow should be prioritized [13]. This prioritization will influence the method of cultivation or filtering techniques. Factors such as gas and small molecule exchange within the device will influence the material of the device and therefore, also the fabrication methods. Material and fabrication are further influenced by the features of the device, which are often application specific.

The application strongly influences the fabrication of a device, both through material properties, such as molecule absorption and optics, and fabrication limitations, such as trapping mechanisms and channel dimensions. When a device is created for either drug discovery or efficacy assays, absorption material properties altering the drug concentration available to cells must be considered. If a device is designed to promote vascularization, variably sized channels will render some fabrication methods unworthy. While base designs may be similar for an assortment of applications, final products can be tremendously diverse. Features for specific applications may conflict with resolution limitations, forcing a design amendment on either the fabrication or material. Additionally, the design will be influenced by downstream processes, as the analysis is not always completed in the device. With organoids, an analysis may not even be the ultimate goal, as the developed organoids may be used in tissue transplantations since the organoid has the same structure and function as the intended tissue [14,15,16].

With so many options for the fabrication of a microfluidic platform, the application must provide priority considerations to drive the manufacturing decision. There are multiple material choices for microfluidic devices, poly(dimethylsiloxane), or more commonly PDMS, is one of the most common in a research laboratory setting. This polymer is commonly poured onto a silicon wafer and cured to fabricate a device [17]. Another method of fabrication is also popular for its rapid prototyping capabilities: 3D printing allows the use of various biocompatible materials giving the user a choice for different material properties [18]. A method that tends to be reserved for finalized designs instead of rapid prototyping is polystyrene (PS) injection into a milled and polished mold [19]. Milling cyclo-olefin copolymer (COC) is an alternative method to PS injection, but is similarly reserved for final design [20]. In some instances, the application demands so much of a material that a new one, such as a special ceramic, is engineered. Occasionally microfluidic principles are implemented outside of a typical device setting, making use of capillaries or standard needles and tubing. Often, the complexity of a design renders one fabrication technique obsolete due to its limitations. However, this has led to the design of microfluidic devices involving multiple materials or fabrication methods. This tends to manifest in layers of a device. 

While it is clear that 3D models have advantages over 2D models, they also have limitations. The stresses induced by blood flow strongly influence the development of organs and tumors and is notably lacking in the standard Matrigel platforms for organoid development [11]. Additionally, results are only acceptable in the scientific community if they are reproducible, which becomes a point of concern when the size of the organoids is not able to be controlled to be consistent in the standard platforms [11]. For these reasons, an improved platform is needed. In this review, we aim to present and evaluate different fabrication methods for microfluidic platforms for organoids, showcasing the advantages and limitations of the different techniques. Since the fabrication is highly influenced by the applications and desired features, we will present some common and cutting edge features as well as highlight the applications that warrant specific or unique designs. Furthermore, some specific applications for pancreatic cancer will be highlighted as well. 

## 2. Applications

### 2.1. Applications Overview

Fabrication options and limitations generate numerous design considerations, and the choice for the final design will be shaped by the features needed for specific applications. Microfluidic devices are being developed to address societal needs such as drug discovery, organ development, and understanding the underlying mechanisms of cancer proliferation. All of these require a unique set of features. Even within these categories, the design requirements vary considerably, as some specific applications require the organoids or spheroids to be developed before introduction into the devices, while others are designed to nurture single cells into fully developed and grown organoids. Some of the applications deal with just organoids, while some will include several different single cell types, depending on the complexity of the model. Additionally, some of the applications will require the possibility of recovering the organoids from the microfluidic device for downstream analysis. With so many variations in applications, it is no wonder that there are so many seemingly slight design variations for microfluidic devices specifically designed for organoids and spheroids. In this section, drug discovery and efficacy, organ development, and general clinical relevance with an emphasis on cancer will be discussed, highlighting the features warranted by the application.

### 2.2. Drug Discovery and Efficacy

In order to assess the specific cytotoxicity of an anticancer peptide (ACP), the benefits of a 3D model, especially with respect to drug resistance, drove the decision to upgrade from the use of 2D models [21]. A 3D matrix provides a container for the drugs and nutrients being provided to the cells as well as the growth factors being released from the cells, creating a more physiologically relevant model [21]. A microfluidic chip was designed to have a central channel to host the hydrogel scaffold in which the organoids grow and a channel on either side of the central one for nutrient exchange [21]. The three channels, with the organoids in a hydrogel in the center, is a common and well-established device for drug evaluation with organoids [22]. The devices allow for live/dead staining directly in the chip, allowing the entire experiment to take place within the device [22]. After treatment, dual labelling was used to evaluate the effect of the drug. A typical three channel device is shown in Figure 1. Instead of the typical solid walls to define the channels, Dhiman et al. designed the chip with hexagonal posts sized and spaced specially to keep the hydrogel and organoids in the central channel and diffusion of nutrients from the side channels (Figure 2) [21]. This design allowed for proliferation of the organoids and the treatment could be easily delivered; without solid walls defining the channels, diffusion of the treatment is less of a concern. Additionally, the organoids could be assessed right in the chip with imaging to evaluate size, growth, and the number of live/dead cells [21].

Since the size of an organoid is used to assess the growth or shrinkage in response to treatment, it is critical that the initial size of organoids is uniform and that the organoids are arranged in a way that each organoid can be identified for final measurements [13]. A microwell in a standard cell culture plate has a greater chance of producing uniformly sized organoids than Matrigel domes; a microfluidic well has even better odds [13]. Even without considering shear stresses and the exchange of nutrients, microfluidic devices are more likely to produce organoids of a consistent size and more representative of in vivo phenotypes [13]. Jung et al. also showed the improved drug screening that can be achieved with a microfluidic device, adding that since a strong level of heterogeneity is preserved in organoids, it is plausible to use devices and organoids to evaluate an individual patient’s resistance to chemotherapy [23].

While the formation of organoids can be a challenge and is critical in its own way, there are instances in which the formation does not drive the design of the chip. Some chips are based on the assumption that the organoids have already been established, and they need to be exposed to the in vitro shear stress and flow conditions as well as treatment [24]. In these instances, the chip can be designed based on a trap. The organoids are flushed through the device and are immobilized by blockades as seen in Figure 3 [24]. Without channels, there is no diffusion, but rather direct application of nutrients and treatment, so the material choice is less restricted, and dosages are better controlled. The organoids remain immobilized as long as there is fluid flow forcing them into the blockade, providing a more representative tumor environment to evaluate drug response [24]. The chip was designed with reservoirs to avoid the need for a syringe pump, allowing the experiment to take place in an incubator [24]. Additionally, the organoids can be subjected to ultrasound in this device, which improves the drug penetration of the organoid [24]. When the organoids need to be removed from the device, reversing the direction of the flow releases them from the trap, allowing this device to be coupled with other applications [24]. 

When evaluating drug response and efficacy, it is critical to have an environment as realistic as possible. The use of organoids certainly elevates the model; however, it does not necessarily perfectly replicate realistic conditions. Several factors may improve the model further, such as introducing fluid flow to mimic the blood flow and inclusion of multiple organs. A microfluidic chip makes this complex system plausible [25]. The use of microchambers connected via microchannels allows for organoids representing multiple organs to interact in a controlled manner [25]. The microchannels ensure that the appropriate nutrients, gas, and metabolites are exchanged, despite any limits from natural diffusion. Additionally, the microfluidic device cannot always provide everything that is needed in the system. Fortunately, it is often relatively easy to couple the device with other systems or methods to bridge the short comings. For example, using the device with a mechanical rocker provides the conditions to mimic recirculation of blood [25]. The PDMS devices created for evaluating the drug response of liver organoids have an easy design that allows for high throughput drug screening, so multiple concentrations of drugs may be tested simultaneously [25]. Often times, the absorption of small molecules by the PDMS is of concern in drug screenings. Jin et al. reported that there was not a statistically significant loss in the concentration of the drug in the device, which is critical to consider when choosing PDMS for drug screening [25].

Typical assays of drugs and biological systems are based on imaging using labeling techniques such as fluorescent staining. For classical MTT assays, a plate reader is used to measure the fluorescence and determine the number of live and dead cells. A problem that is not encountered in traditional 2D models is dye penetration to all cells of the organoid regardless of position in 3D space, i.e., outer and inner layers. To address this, Ermis et al. developed a microfluidic device that can facilitate staining of cells during organoid formation. This allows the dye to reach all layers of the organoid and produces a more prominent fluorescence signal as opposed to staining after the organoid formation [26]. Finally, this device allows for a dye to be applied and flushed in a more consistent manner to all organoids than would be possible in a standard 3D matrix platform [26].

Microfluidic devices like these have surmounted many of the limitations of traditional label-based assays. Label-free imaging techniques such as Raman spectroscopy [27,28], impedance cytometry [29], selective plane illumination microscopy [30], or brightfield imaging [31] can also be used to perform organoids-based evaluations. Despite their utility, there are some limitations that should be kept in mind for each technique. For example, both brightfield imaging and impedance cytometry do not lay a foundation for staining for specific expressions of the organoids [29,31]. On the other hand, Raman spectroscopy-based techniques need more standardization across laboratories and different experiments [27,28].

Drug screening is not the only purpose for which controlled fluid flow is critical. The nervous system is quite complex, including complex fluid interactions. While most organs require the consideration of only one fluid, such as blood, brain organoids require the consideration of cerebral spinal fluid (CSF) as well as blood [32]. The boundaries between the CSF and blood brain barrier as well as CSF and interstitial space present an interesting condition in which there exists bi-directional flow [32]. The bi-directional flow is critical to the development of several neuronal features, making it essential to recreate this attribute when studying the development of the nervous system with organoids (Figure 4).

### 2.3. Developmental Biology

Aside from understanding the natural development of complex organs and systems, it is imperative to understand how this development is affected by variations in the developmental environment. Brain organoids will generally require the formation of embryonic bodies from human pluripotent stem cells (hPSCs) before maturing into organoids, which have a possibility of merging and growing so large that necrosis and hypoxic cores are induced [33]. Additionally, the different stages of this development has required switching the organoids to various platforms, which can harm the organoids by both physical damage and loss of cells as well as introducing contaminants [33]. A microfluidic device can be tailored to address these issues. The size of the microwells helps to isolate the organoids from each other as well as physically constraining the size. This, combined with forcing an open air interface, reduces the necrosis and hypoxic core [33]. Incorporating a mesh membrane into the well (Figure 5) will elevate the organoid, allowing media on the bottom to be changed without damaging the organoid [33]. Furthermore, the mesh membrane can be used for the formation of embryonic bodies as well as the maturation of the organoid, eliminating the need to transfer to different platforms [33]. Furthermore, the mesh membrane can be used for the formation of embryonic bodies as well as the maturation of the organoid, eliminating the need to transfer to different platforms. On top of the membrane, along with the cells, the media can be easily changed from neural induction and expansion media to Matrigel, and, finally, media can be eliminated allowing for the air interface. This design allows for the organoid to form and mature, and the media at the bottom may be treated to test for developmental conditions, such as prenatal exposure to cannabis [33]. A porous membrane that suspends the cells above a reservoir can be modified to eliminate the wells shown in Figure 5. Haque et al. developed a microfluidic device that has a large chamber above a porous membrane instead of the wells so pancreatic ductal adenocarcinoma (PDAC) PDOs could be cultured with stromal cells to mimic the interactions characteristic of the tumor microenvironment [34].

Human brain organoids are not the only ones created from hPSCs. While human islets are quite complex, especially when it comes to functionality, there are promising results regarding the use of organoids as an in vitro model, which will open many doors in the area of diabetes and artificial pancreas research [35]. The same concept of the microwells with a mesh membrane elevating the organoids remains from the previously mentioned design; however, blood flow is a critical component for the appropriate signaling to properly form functioning islets and is therefore a critical component for the formation of islet organoids [35]. This is accommodated by taking advantage of creating layers in a microfluidic device. A channel with a width nearly the same as the device allows the flow of media through the entire bottom of the device [35]. Instead of having membranes in each well, a mesh sheet is placed just above the media channel [35]. To house the organoids, a PDMS sheet with holes through the thickness is seated above the mesh sheet instead of creating individual wells [35]. The theory of the device is quite similar to the previously described design and the features can be seen in Figure 5, with the addition of fluid flow to favor the biological formation of islets [35].

In the interest of tissue engineering and regenerative medicine, one of the most important factors is the vascularization of the tissue. When the experiments get scaled down to the microfluidic level, vascularization remains a pivotal factor, but becomes difficult to achieve. The design of the microfluidic device can induce fluid flows that are physiologically integral in the development of a vascular network [20]. By trapping the organoid with flow on two sides, instead of just one that is more common, a vascular network can be achieved, with interstitial flows and stresses comparable to human capillaries. This is accomplished with a relatively simple U-trap design. A U-shaped channel houses the flowing media and parallel sections of the channel are connected with a smaller, perpendicular channel, which will trap the organoid [20]. The only drawback for this design is that it relies on the usage of previously established organoids, as the organoid does not mature in this trap [20]. It is possible that not all of the organoids formed outside of this device will fit into the trap if the size is not regulated during organoid formation [20].

### 2.4. Oncology and Clinical Applications

Tissue engineering and regenerative medicine have been becoming more prominent over the last several years; however, cancer has been a target for researchers for many years. For this reason, it is critical to have a well-developed approach to forming consistent, reliable organoids and be able to couple this formation with additional downstream manipulation. Most of the previously described microfluidic devices have been designed for a specific assay or imaging technique, and that is all that can be done with the organoid in the chip. However, it would be significantly more useful if a device can release the organoids. Taking advantage of reversible bonding can help make a device suitable for producing organoids to be used outside of the device [36]. To achieve this, there should be a combination of features. Pinho et al. developed a device in which the bottom layer of the device has several cylinders to act as the wells for spheroid formation and a top layer has an inlet, outlet, and channels to each of the wells for continuous profusion of media to the spheroids [36]. When the two layers are reversibly bonded, the formed spheroids or organoids may be removed for use in assays outside of the device [36]. Since media is pumped through the device, it is possible to treat the organoids in the formation chamber; however, more complex analysis may not be possible on chip [36]. This device, while simple, showed greater cell viability than standard culturing indicating this is a better model to use for exploring the mechanisms of cancer [36].

As organoids and spheroids become the preferred model to use for understanding the underlying mechanisms of cancer, the methods used to analyze become the limitations. Imaging is the standard for molecular distributions; however, some of the structures of the spheroids may interfere with the light, obstructing the image [37]. Just as with tissues, it is possible to obtain the morphological information of structures within the spheroid through sectioning. However, since spheroids are notably smaller than most tissue specimen, sectioning a spheroid seems to be quite an unlikely task [37]. There is the potential to section the spheroid in a microfluidic device, although there are multiple factors that must be considered, including the material properties of the device to accommodate slicing [37]. It is, once again, ideal to use temporary bonding to create a layered device in order to easily remove the top of the device. An epoxy-based material used for the bottom of the device has channels load media and organoids and trenches in which the organoids become trapped [37]. The removable top layer mainly serves as a lid and has inlets and outlets for the media [37]. Once the organoids are trapped in the trenches, the inlet is used to introduce paraffin into the device, which once dehydrated, will encapsulate the organoid [37]. The top of the device can be removed, and a blade can slice through the paraffin and epoxy-based bottom to produce a slice of the organoid [37]. Having access to the morphological features of the organoid will also show the morphological features of the tumor and can be used to not only understand cancer development but also treatments [37].

When studying cancer, it is important to study the progression of the cancer, including metastasis and migration of the cells. Cancer stem-like cells (CSCs) can be used to evaluate the progression as well as therapy resistance, and the microenvironment can be tuned such that the CSCs are enhanced to mimic in vivo conditions. Since CSCs display stem cell properties, it is believed that treatments should target these cells [38,39,40]. Unfortunately, they are challenging to study, which drives the innovation in the assays. Encapsulation of the cells in either a classic collagen scaffold [38] or hydrogel microcapsule [39] presents the same limitations of developing organoids in a hydrogel matrix. A microfluidic device can be designed to aid the formation of spheroids from CSCs with a precisely tuned microenvironment [41]. Chen et al. developed a device that has a serpentine channel and microwells for trapping and formation [40].

Alternatively, U-shaped obstacles can be used to trap (Figure 6) the cells instead of microwells, exposing the cells to higher levels of glucose and oxygen, which will enhance the CSCs. To assist in overcoming the challenges of loading the cells into the traps, the device is mounted to a plate at a 45° angle such that gravity will keep the cells in the trap [41]. Additionally, in the microfluidic device, the flow rate can be finely tuned to expose the cells to the appropriate shear stress [41]. The fluid flow can be simulated with the design of the device, minimizing the amount of experimental optimization [41]. With the ability to alter various parameters, it is possible to evaluate the effect of each on overall cancer progression [41].

This isolation is imperative when working with human tissue. In many of the previous examples, organoids were formed from stem cells, however, it is also possible to generate organoids from resected tumor tissue samples [42,43,44,45,46]. Tumor tissue is used as a source in organoid and spheroid formation in multiple cancers, both to focus on one type of cell and to utilize a variety of cell types to most accurately recapitulate the native tumor environment [43,44,45,46]. Additionally, the spheroids of the primary tumor need to be isolated from the other cells. The spheroids from the tumor tissue will retain the morphological and cellular features of the tumor itself, making these spheroids an excellent source when studying cancer [42]. When using tumor tissue, it is possible to have some undesired cell types in culture. In this case, the spheroids of the primary tumor need to be isolated from the other cells.

Contrary to using the trapping method to only form the organoids, a similar trap may be used to isolate organoids of a specific size. Instead of the obstacle being fully closed on one side, they can be open to allow single cells and small organoids to pass through, as shown in Figure 6 [42]. The open end of the trap not only allows single cells to pass through, it also allows a reverse fluid flow to dislodge the organoids in the trap, releasing them to be used in further analysis, even outside of the device [42].

Alternatively, a spheroid or organoid may be trapped to expose it to specific conditions. A trapping site in a microfluidic device may just be channels that are designed to be too small for a spheroid to pass through but large enough for other media (Figure 7) [47]. Separate channels on either side of the spheroid allow it to be exposed to two different chemical gradients, with a small area for a hydrogel matrix in between the media channels and the trapped spheroid for the growth of the sprouts [47]. With this design, tumor angiogenesis can be studied with various chemical gradients.

There are various sources for cells that are used in organoid and spheroid formation, most commonly stem cells or tissue. Other cells that can be considered for organoid or spheroid formation are circulating tumor cells (CTCs). CTCs are often studied to determine cancer progression and metastasis, since they migrate from the original tumor site towards other organs [48]. These cells are rarely considered for spheroid formation despite their wealth of information. This could be attributed to the fact that they tend to be rare among blood cells, making it difficult to evaluate them [48]. In this circumstance, it is necessary to design the device to trap single cells instead of trapping spheroids. A microfluidic device can be used to isolate the CTCs from the blood cells and culture them into spheroids for analysis [48]. Blood cells and CTCs are different sizes and have different properties, so when an acoustically induced oscillation of an air bubble in the device provides an additional force on the cells, they react differently, ultimately separating as the blood cells continue in the main flow while the CTCs get trapped in a rotational flow at the bubble [48]. The microfluidic device also has a chamber in which the spheroid can grow, with media flowing through the main channel after the blood has been washed from the device [48].

### 2.5. Applications Discussion

The development of the human nervous system and testing the efficacy of drugs are two very different implementations of organoids and spheroids and, as such, warrant very different designs of the microfluidic devices used. Some features will remain the same, such as the common requirement for flow, both to deliver fresh media to the cells and to create the appropriate shear stresses to mimic in vivo conditions. Additionally, fluid flow can provide the advantage of separating different cell types to isolate either organoids from single cells or differentially sized single cells to develop spheroids. When trapping organoids, the design of the trap may differ greatly depending on the actual application and to what conditions the organoids need to be exposed.

Another major factor that must be considered when designing a microfluidic chip for a specific application is what processes are coupled with the device. If the organoids need to be removed from the device, the features may be altered to accommodate this. The features may also differ if the organoids need to be assessed individually or as a group. Some applications may work best with organoids of a specific size, and the device will either need be designed to grow the organoids to an appropriate size or filter for it. While the specific applications may favor new microfluidic designs to accommodate features, there is still the consideration for mass-produced devices and adapting platforms. Geyer et al. established a PDAC co-culture model in a commercially available microfluidic platform and successfully created a specific microenvironment in a generic device [49].

## 3. Device Fabrication

### 3.1. Fabrication Overview

There are extensive options for the fabrication of a microfluidic device suitable for patient derived organoid models in terms of substrate used and manufacturing methods. The materials and manufacturing methods both have individual limitations that must be considered when designing a chip. Material choice is often driven by the needs of the sample and goals of the application. A strong deciding factor is the transparency of at least a section of the device to allow imaging which may be necessary to monitor the formation of organoids or the effect of drug treatment. This would promote the choice of a clear substrate. Alternatively, material may also be chosen based primarily on the permeability that allows gas exchange. This permeability can be a debilitating property though, absorbing molecules for which the concentration can alter the results of the experiment, leading some laboratories to choose a different material for the device.

A material choice may force an alternate fabrication method; some polymers may be cured in a mold, while other substrates can be 3D printed. Three-dimensional printing greatly broadens the material choices while still promoting rapid prototyping. While PDMS and 3D printing are among the most common methods to produce a chip, they are not the only options. Common lab supplies such as capillaries can sometimes be repurposed to achieve the same effects as a microfluidic device. Alternatively, solid blocks of material, often also polymers, can be micro milled to the desired design.

Often, one method or manufacturing technique is not sufficient to meet the needs of the sample and the limitations of manufacturing. Limitations in resolution, overall size, and mass -production capabilities may disqualify any of the methods as a viable option. In these cases, various materials or methods are combined to create the microfluidic device. In this section, the most common substrates and manufacturing methods will be discussed in detail, along with their advantages, limitations, and applications in cancer research.

### 3.2. Soft Lithography—PDMS

There are two major contributors to soft lithography, especially PDMS, as the first choice for microfluidic devices in research laboratories: fabrication and material properties. The desirable material properties include transparency for ease of imaging, permeability, which allows the exchange of gases while not absorbing water, and being inert and nontoxic, making it safe for cells [20]. The permeability does cause an issue in some applications, as this material is also permeable to non-polar organic molecules; therefore, PDMS should be considered with caution [20]. The fabrication, detailed by McDonald et al. [17] and summarized in Figure 8, begins with creating the design in computer-aided drafting (CAD) software. This design is then translated in the software to a negative mold design. The physical mold is created by applying a photoresist layer to a silicon wafer and treating it with light. The device itself is created by curing a 10:1 ratio of base and curing agent in a vacuum. The device can be cut into the appropriate size with a precision knife, and holes may be punched out and subsequently reversibly attached to a glass slide with heat. The final device can be made to a resolution of several nanometers in all axes; however, care must be taken to ensure that the width-to-height ratio of the channels is sufficient to avoid collapse [17].

The fabrication of a PDMS chip is relatively simple and requires minimal training, so platforms with a simple design are convenient and achievable in most laboratories. A simple design does not mean that the chip cannot resolve challenges with imperative functions. Bradney et al. demonstrated how a very simple PDMS device with essentially only one channel between an inlet and outlet can be instrumental in studying PDAC [50]. With two layers each containing a channel bonded together so that the channels combine to form a cylindrical tunnel through the device, an epithelial cancer cell duct can be created to imitate the tumor microenvironment [50]. Another relatively simple design is a T-junction, shown in Figure 9, which is traditionally used to form uniform organoids by systemic encapsulation of a consistent number of cells in a matrix phase to create size homogenous organoids. The T junction also serves as a building point for other designs. One modification is to create a cross junction from the T junction by splitting an inlet channel, as seen in Figure 10. In this case, the matrix phase is split such that the cells, which flow perpendicular to the matrix phase, are met with oil from either side, completing the encapsulation [51]. Alternatively to splitting one channel, it may be necessary to have a secondary aqueous phase after the cells before the matrix phase. As this design modification adds only one channel that is parallel to the oil phase channel and perpendicular to the aqueous cell phase, it remains a simple, elegant design to encapsulate cells for organoid culture that is easily created with PDMS [52]. This method controls the number of cells per droplet, thereby influencing the final size of the organoid.

While the T junction may be one of the simplest designs for a microfluidic platform design specifically for 3D cell models, it is certainly not the only option. There are recent developments in the design of microwells for the formation of the 3D cell clusters. It is possible to advance past the limitations, such as fluid flow for nutrient exchange and shear stresses, attributed to classical microwells in a plate with microwells in a microfluidic device [53]. The microwells are designed as essentially small channels which are horizontal instead of vertical, removing the need for micro milling and allowing this device to be made of PDMS in a standard manufacturing process. The only modification is that the wells are loaded with centrifugal force instead of gravity [53].

Gravity can be key in the formation of organoids in a process known as the hanging droplet method, where spheroid formation can be achieved in a standard culture plate without microfluidics [54]. This method takes advantage of the adhesive, viscous properties of the suspension substrate, which when coupled with the downward pull of gravity, creates a suspended droplet that houses the cells. Similar to the encapsulation methods utilizing a T junction, the hanging drop method can be achieved with a basic PDMS device [55]. Frey et al. developed a PDMS device that is inverted when used to take advantage of gravity and create the drop. The key features of this device are a well that becomes inverted when in use and a hydrophobic rim that will force the droplet shape, creating better control over the droplet and subsequently the formation of the organoid.

Similar to using the microchannels as microwells, the PDMS channels can be designed such that they create traps and small passageways to filter rather than form organoids [56]. The tolerances allowed with PDMS allow for small passageways filled with a scaffold gel to be placed near a small piece of material that would trap a single organoid. With the design of additional channels, flow is established around an organoid and it can vascularize through the gel and into the channels, creating a representative microenvironment [56].

The trapping mechanisms do not need to be complex as demonstrated by Zhou et al., who successfully created a PDMS device that utilizes simple obstructions to form triangular trapping sites similar to the trapping described in Figure 6 [42]. While many microfluidic devices are focused on the formation and culture of organoids or spheroids, this device is intended to be used with previously established spheroids and serves as a mean to filter unwanted single cells or improperly sized spheroids out, isolating the targeted spheroids [42]. The design of the inlets, outlets, and channels make it simple to reverse the flow within the device to be able to harvest the trapped spheroids [42]. The ease and low cost of manufacturing makes it feasible to modify the size of the trapping sites, making this device applicable to various spheroids and applications.

The major benefit of trapping cells or spheroids in a location is to provide flow around the cells, providing a representative nutrient and waste exchange. However, some large barriers may overcomplicate the design or possibly hinder other features of the device. It is possible to trap a hydrogel containing cells by manipulating the physical properties. The addition of well-placed small ridges to the channels of a device will alter the contact angles of the droplet, therefore increasing the wettability, which will prevent the droplet from moving when the flow of media is introduced to the device [57]. The size of the ridges remains within the capability range of PDMS, avoiding complex manufacturing. Additionally, the surface properties of PDMS can be manipulated such that it can be completely hydrophobic, further optimizing the contact angles of the droplet [57].

There are further options for trapping cells in a specified location. To move particles and cells, surface acoustic waves (SAW) can be used, a technology that was first developed to be used with silicon and glass devices. However, Guo et al. adapted the mechanisms for PDMS devices, significantly increasing the feasibility to use this device in rapid prototyping [58]. Chen et al. then further developed this technology to not only make part of the device reusable but also applicable to organoid and spheroid formation [59]. A PDMS chip with channels is set on a wave generator with a thin layer of oil; cells are loaded into the channels, and then the applied wave guides the cells into clusters at the nodes of the wave, essentially trapping the cells for formation without physical barriers in the device [59]. This eliminates the need for complicated designs in an effort to reverse the fluid flow to harvest the organoids, and as the cells are suspended, no additional coatings are needed to prevent the cells from adhering to the PDMS.

While PDMS is often an ideal choice due to the ease of fabrication, it is possible to deviate from the standard fabrication to achieve more complex design features. The microwell is rapidly becoming a standard design for chips designed for organoid formation. However, there is generally fluid flow only at the top of the well, if there is flow at all. The standard design is improved by adding flow in the form of media profusion at the bottom of the well in addition to at the top. This certainly complicates the design of the device, but if the theory behind each feature is preserved, each feature can be accomplished with individual layers, and this break down of design allows the device to be fabricated with PDMS [35]. The device needs a top and bottom to contain the media and organoids, as well as provide inlets and outlets for the exchange of media and introducing the cells. In order to form the microwells, a layer of PDMS with punched holes is added [35]. While this layer traps the cells, by itself, it does little to elevate the design and function of the device. However, coupled with a porous membrane on the bottom of the microwell layer, the organoid becomes suspended above the bottom of the device, allowing media to flow both at the top (through channels) and the bottom of the wells [35]. PDMS definitely has a place in the research laboratory; however, concerns regarding molecule absorption and the ability to create wells may preclude it for specific applications. Additionally, using PDMS creates a closed system, which will hinder downstream capabilities and also restrict the possibility of using it for some applications.

### 3.3. 3D Printing

The advances in 3D printing allow the use of many substrates in prototyping, including microscale chip manufacturing. One of the advantages of 3D printing is the material options as many materials are capable of being printed. While these options translate to increased time and effort on optimization of a balance of printability and biocompatibility [18], it also eliminates some of the previous limitations. Figure 11 summarizes the general process of 3D printing a microfluidic device, which also begins with design in CAD software. This is where the similarities with PDMS production end, as 3D printing utilizes a particular printer to create the final device and eliminates the need for a negative mold.

The previously mentioned devices are completely enclosed systems; however, this is not the only option. Using 3D printing, it is possible to create an open device [18]. One of the major benefits of keeping the device open is to be able to place and then remove an organoid, even after it has vascularized within the device [18]. This leaves a great deal of potential for automation downstream [18]. Salmon et al. created a device that allows two channels to flow around an organoid to promote vascularization [18]. The device was printed with Dental SG for biocompatibility and since the device is open on the top, optical properties were not critical for imaging [18]. While imaging concerns did not contribute to material choice, the ability to print to the desired resolution did, as some materials could not form the small channels [18]. Even though the material is biocompatible, a washing process still had to be optimized to remove any unreacted polymers [18].

Three dimensional printing unlocks many possibilities in both design and scalability. When a 3D printed device has an opening at the top of a well, not only can an organoid be easily removed but it can also be easily imaged, allowing these devices to compete in the same league as PDMS devices. Imaging is a critical form of evaluation for organoids, so it is reasonable that devices may be designed to optimize imaging capability. One of the challenges associated with growing organoids is the detachment of the Matrigel from the bottom of the well or plate. By adding notch features on the walls of the well, the phenomenon of detachment is minimized [60]. Khan et al. developed a 3D printed design that minimized the detachment of Matrigel with the addition of the notches, a feature not possible with PDMS [60]. One disadvantage shared by PDMS and 3D printing is the scalability to mass production.

### 3.4. Injection and Milling

While PDMS and 3D printed devices have proven to be an ideal solution for fast cost-effective prototyping and proof of concept scenarios, they are not always applicable outside of a research lab due to both material properties and manufacturing limitations [20,61,62,63]. There are alternative materials that are accompanied by alternative fabrication methods. There are two classifications of manufacturing, additive or removal of material, and either is a suitable option. One option is to use a mold, generally made from an alloy that is polished to reduce the surface roughness, and inject PS [19,20] Prototypes for an injection molded device can be 3D printed before large scale production of the final product [19]. Additionally, the surfaces of the chip can be treated to achieve optimal surface properties [19]. A device was fabricated with PS injection molding to create a patterned cell culture [19]. Initially, this was for 2D cell lines, however, the design was modified to be able to accommodate spheroids [19,20]. Ko et al. designed a device with a rail within a well that is placed above the substrate at a height optimal for taking advantage of capillary forces [20]. The rail has a hole perfectly sized to accommodate a 200 µL pipette tip to load the spheroids, and the capillary forces disperse the cells under the rail and form a concave interface between the rail and substrate, an ideal surface for integrating with media within the well on either side of the rail [20]. This device provides conditions for various assays and is designed with the same specifications as a 96-well plate, allowing for automation or coupling with a wide array of analytical techniques [20]. Tung et al. took a different approach to the hanging drop method previously described and created a PS injected plate that consisted of injection sites to accommodate a pipette tip, the ridge to form the drop, and a bottom well for water in order to control the humidity of the environment [64]. Polystyrene provides the option to inject into a mold or mill a sheet of it. Rodopulo et al. opted for layers of milled PS to create a microfluidic device for the hanging drop method instead of injecting into a device. The layered design allowed for features that could accommodate co-culture [65]. Similar to 3D printing, injection molding offers a selection of materials. Schütte et al. used injection molded cyclo-olefin polymer (COP) with diodes to create an electrical field that would guide the cells to the designated formation zones [66]. The device was first designed for and tested with hepatocytes and endothelial cells [66] and was later proven to also be successful with PDAC cells [67]. One of the interesting notes from this device is that it is more of a bridge between 2D and 3D models; as for the PDAC lines, the cells were 2D cell lines but showed the morphology characteristics of 3D spheroids [67].

An alternative to additive manufacturing with PS is milling a block of COC to the desired specifications [63]. The optical properties of COC make it a contender for such devices, allowing for imaging which is often imperative for organoid work [63]. Similar to PS, COC also does not interfere with the chemicals involved in culture and drug screening, adding to the attractiveness of COC. The surface properties combined with the robustness of the material and the ability to manufacture on a mass scale make COC an ideal option. Quintard et al. designed a device that would promote vascularization of previously established tumor organoids [63]. Once the design was optimized specifically for the targeted organoids, the devices could be mass produced. Arguably, one of the biggest advantages of this design is the potential for automation. Each channel is designed with one inlet and one outlet, housing one organoid. While this design does not allow for cell-cell interactions, it does offer absolute control over the flow rates that each organoid is exposed to with a syringe pump [63]. Due to the time commitment for organoids, any automation serves as an advantage.

### 3.5. Other—Repurposed and Special Materials

The design of a microfluidic device is dependent on the end goal, and sometimes the end goal may just be preparation for a future step, which may diminish the motivation to design and fabricate entire new devices for one step in the process. For example, applications such as bioprinting used for tissue engineering require consistently sized organoids to be uniformly coated with a hydrogel. Mesquita et al. developed a microfluidic device that utilizes glass capillaries to achieve a predictable, reproducible hydrogel coating on the spheroids [68]. In this case, assembly rather than manufacturing is required. The glass capillaries used are commercially available. The use of available features makes the design and fabrication processes easier and less time consuming, which makes the solution ideal for prototyping and proof of concept. The previously mentioned PDMS device using SAW to generate organoids can be modified to utilize glass capillaries [69]. This elevates the device from being ideal for prototyping to being a candidate for large scale use. The ingenuity of the design is in the wave generator and not in the full device design, allowing the PDMS device to be replaced with glass capillaries [69]. The use of glass channels expands the potential uses with the surface properties of glass being more suitable to avoid chemical absorption and transmit energy when compared to PDMS.

As previously mentioned, some devices are designed to encapsulate the cells in a hydrogel to promote the formation of organoids. In its simplest platform, this may not need to be a conventional standalone device but rather an assembly. Shao et al. optimized a platform to use a standard, commercially available transparent silicon tube to serve as the channel to deliver the continuous oil phase. A commercially available sharp mouth needle is inserted into the tube via a puncture and delivers the cells in a gelatin methacrylate (GelMA) mixture as shown in Figure 12 [70]. The flow rates are controlled and optimized to form the ideal size gel spheres. As the spheres continue to flow through the tube, they are cross-linked while passing under a section of blue light [70]. This platform brings together commercially available materials to create a simple, cost effective, and easy to fabricate method of encapsulating the cells for spheroid formation. The limitation with this method is that organoids must go somewhere for the assays and analysis, so this must be coupled with another process.

It can be beneficial to have both 3D and 2D cells for an experiment to either better understand the differences between the models or validate the 3D model. However, in order for the results to be comparable, the cells in each model must be subject to the exact same culture conditions, and one way to achieve this is through culturing them together. The challenge to culturing these cells together is that they require different conditions, even different surface conditions—the 2D cells need an adherent surface while the 3D cells require a suspension surface [71]. The different surface conditions pose an interesting dilemma, one that is solved with a microfluidic device created by Järvinen et al. and made of relatively easily manipulated materials. Ormocomp is a ceramic that is naturally cell adherent, which provides the surface for the 2D cell lines, and plasma exposure renders the surface cell repellent which provides the surface for the 3D cell lines [71]. Applying a film to the desired surfaces during fabrication controls the exposure to plasma and therefore controls which surfaces become repellent and which remain adherent [71]. The material is fabricated to create microwells to trap the cells destined for 3D culture and chambers for the 2D cells [71]. One of the limitations of using a ceramic is that it is not gas permeable, meaning that oxygen cannot get into a completely ceramic device. The device is sealed with PDMS, which is gas permeable, to overcome this issue [71]. The ability to culture 3D cells in the same conditions along 2D cells will help to validate studies and better understand the 3D environment; however, limitations regarding the capacity to work with ceramics and plasma treatment may occur.

### 3.6. Combination of Fabrication Materials and Methods

Sometimes the need for multiple materials stems from the fabrication needs, not necessarily the needs of the cells. Studies have shown that a concave well (Figure 13) is more successful at creating uniform organoids than a traditionally flat bottomed well (Figure 13) [72] however, this design is difficult to fabricate with PDMS. Interestingly, concavity is considered a disadvantage in most microfluidic platforms and is generally an area of concern for using wet etching techniques to fabricate wells and channels on glass [72]. Sun et al. used this “disadvantage” to easily fabricate concave microwells on glass and bonded a PDMS piece with channels on top of the glass to complete the chip. With the wet etching techniques, the wells were made deep enough that the spheroids would not be adversely affected by the shear stresses induced with the flow of media through the device [72].

Other fabrication limitations may not directly change the material of the device itself but may still warrant a modification to the standard fabrication procedure. While PDMS chips are typically fabricated from a wafer, this eliminates the possibility of having microwells in the device to trap cells for organoid or spheroid formation. The wafer can be easily replaced with a mold, which is what Behroodi et al. did. For optimal spheroid formation, the microwells are conical in shape, with a 25° angle, and are 300 µm deep [73]. In order to achieve these dimensions on the mold, a 3D printer capable of resolution on the microscale was used; however, this printer had a maximum chip size of 7.7 mm × 4.8 mm, which is insufficient for creating the entire chip, designed to have final dimensions of 30 mm × 15 mm to include sufficient channel length [73]. In order to curb the limitations of the printer, the group combined two different techniques to create the mold, adding the 3D printed segment into a larger chip that was CNC milled [73]. In the end, the PDMS chip was cast with a mold that was both 3D printed and micro-milled.

It is reasonable to assume that as the purposes and goals of a microfluidic device increase in complexity, so will the limitations of fabrication, creating a demand for combinations of materials and fabrication methods, as demonstrated above. In an effort to mimic the complexity and mutual dependence of systems in the human body, a platform was created to host up to six different cell types [74]. The complexity of this design renders a typical PDMS microfluidic device inadequate. The device was made of polymethyl methacrylate (PMMA) layers that were laser cut and a polycarbonate membrane [74]. The layers were bound together with double sided tape and finally mounted on a glass slide. These materials allowed for imaging and limited the drug and protein absorption into the walls of the device [74]. The device was designed for observation of the systems supporting each other, especially for drug response; therefore, it was not important to optimize for organoid formation, as the device uses established organoids strategically placed within the layers [74].

### 3.7. Fabrication Discussion

With so many options available for creating a microfluidic device, it is imperative to consider the limitations of each and the priorities of the application. A summary with the primary advantages and cautionary limitations is provided in Table 1. With PDMS, the gas exchange allows oxygen but also potentially allows drugs to permeate, and it is easy to produce in a lab but does not scale well. In regard to 3D printing, the resolution allows for specific features, and there is a relatively broad material choice. However, care must be taken to consider all material properties, especially optical ones for imaging. If organoids are to be the new gold standard, large-scale production of the devices is a prerequisite, and injection molding and milling are both superior to PDMS and 3D printing in that sense. Assuming that large scale production is not a concern during design, using common lab supplies to achieve the microfluidic properties may be the best option for a proof of concept even before the rapid prototyping stage. With a variety of options available, prioritizing the features may result in using a combination of materials to make the device resulting in layers. Microfluidic devices are constantly being designed to accommodate features driven by the application, some of which were discussed in the Section 2.

## 4. Conclusions

Plenty of microfluidic devices currently exist for studies involving organoids, providing both material and feature options. However, choosing a device that exists without any modifications may not be the best option. There is not a single design that is applicable to every application. The small molecule exchange and imaging requirements of the application will strongly drive the material and fabrication preferences. Distinct applications require specialized features which often compel the researcher to design a new device, and these features will undoubtedly influence material and fabrication selection. A device may be a combination of features found in existing devices, yet the combination may be unique, demanding a new device design. It is important to consider all previous designs to simplify the designs of a new device by using modifications or combinations of previous devices. It is critical to begin with the application priorities, including downstream automation and process coupling, to avoid any issues with feature or material limitations.

## Figures and Tables

**Figure 1 ijms-24-10882-f001:**
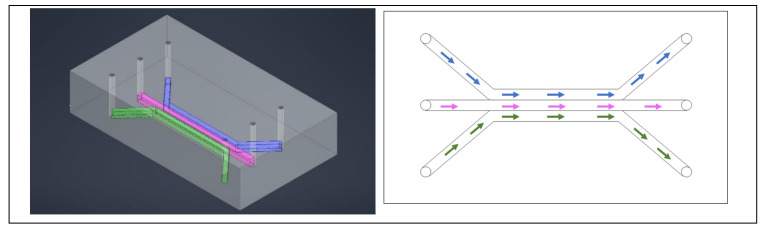
A typical three channel device, with three inlets and three outlets. The colors help identify individual channels. Cells are typically in the center channel, allowing media, drugs, and stains to be perfused through the outer two channels. The left image is a simulated model of the device and on the right is a drawing of a comparable three channel device.

**Figure 2 ijms-24-10882-f002:**
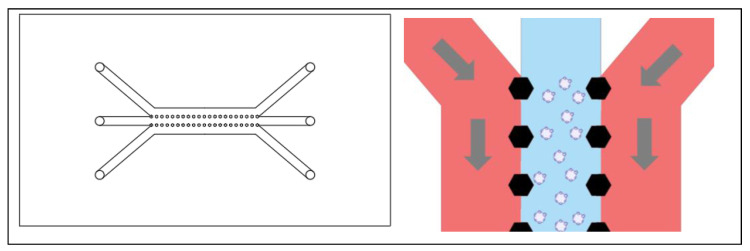
**Left**: A three channel device with hexagonal posts instead of solid walls to enclose each channel. **Right**: A close up showing the shape of the hexagonal posts supporting a hydrogel in the center channel.

**Figure 3 ijms-24-10882-f003:**
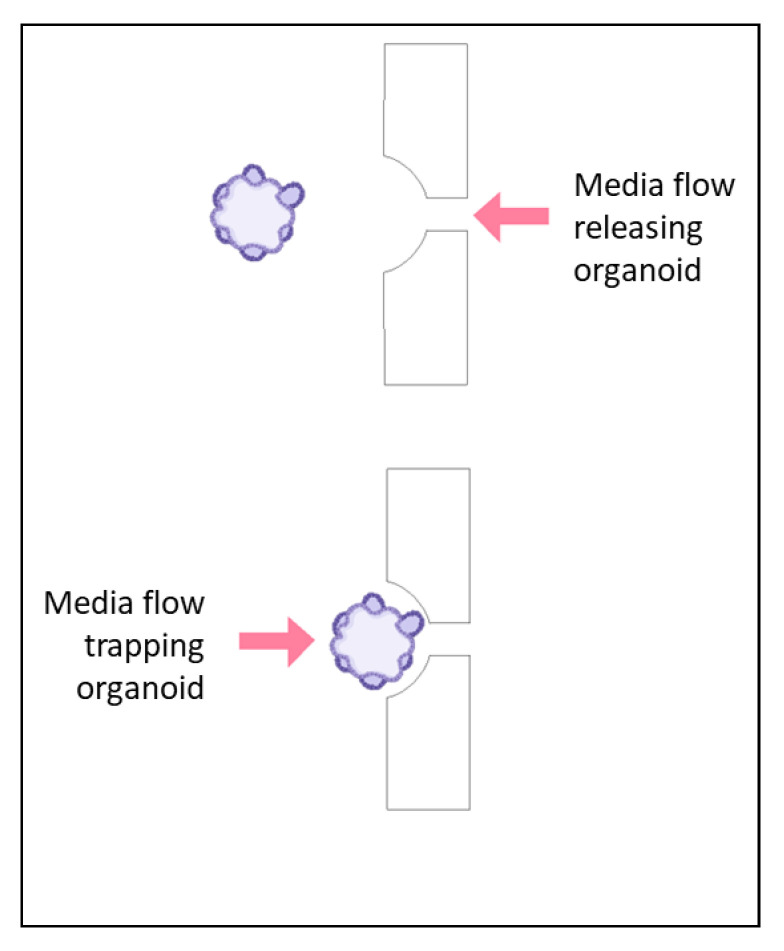
An organoid will be either trapped or released from a trapping blockade depending on the direction of the flow of media, as indicated with the arrows. In the bottom scenario, the organoid is trapped in a rectangular blockade when the media is flowing from the direction of the large opening of the blockade toward the smaller opening. When the flow is reversed, as shown in the top scenario, the organoid is released from the blockade.

**Figure 4 ijms-24-10882-f004:**
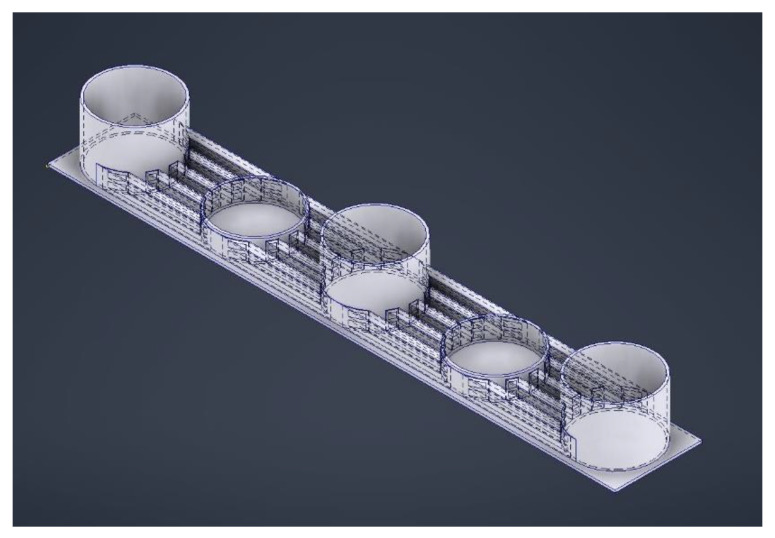
Microfluidic device with five wells: the deeper ones to serve as media reservoirs and the shallower ones to house the organoids. The channels connecting the wells are layered to provide enough flow for the exchange of nutrients and waste while limiting the shear stresses for optimal growth of the brain organoids.

**Figure 5 ijms-24-10882-f005:**
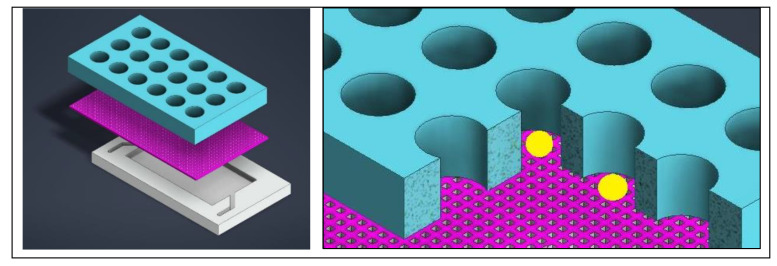
Regardless of the full device layout, the bottom should provide a shallow well for media, with a mesh above it so that the media can get into the larger organoid wells that are on the top. The mesh prevents the organoid from sitting on the bottom of a well and allows for volume control. On the right, organoids represented by yellow spheres are shown resting on the mesh in the well.

**Figure 6 ijms-24-10882-f006:**
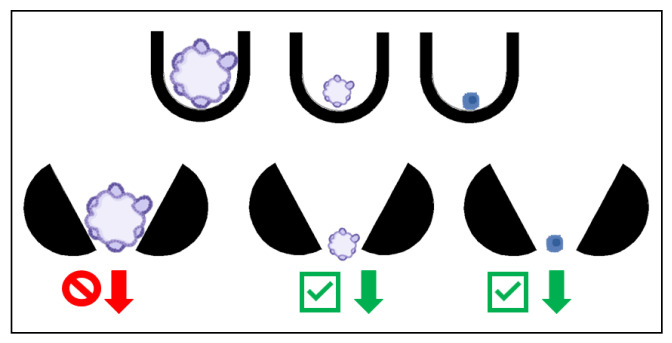
The U-shaped traps on the top row will trap organoids of any size (left two U traps), as well as single cells (right U trap). The open traps on the bottom row allow the smaller organoids and single cells to pass as indicated with the green check mark arrows and prohibiting the larger organoids to pass as indicated with the red forbidden arrow, acting as a filter for organoids of a specified size.

**Figure 7 ijms-24-10882-f007:**
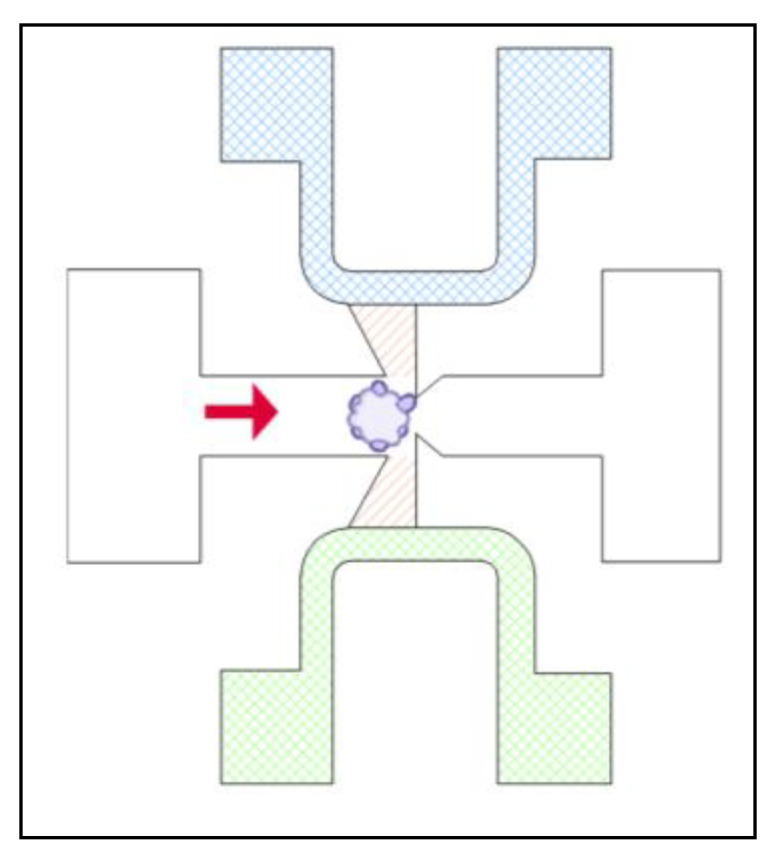
Microfluidic device designed to trap the organoid with media flow (indicated with solid red arrow), provide a hydrogel matrix (striped, orange) and expose the organoid to two different sources of media (cross hatched green and blue) to expose opposing sides of the organoid to different chemical gradients.

**Figure 8 ijms-24-10882-f008:**
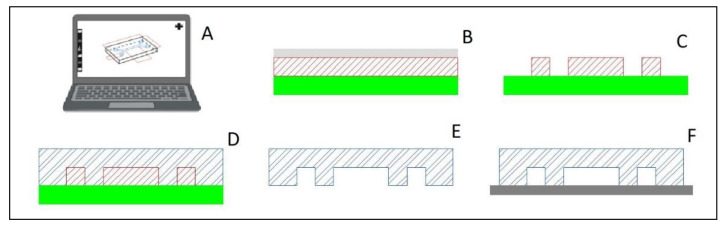
Basic manufacturing process of Poly(dimethylsiloxane) (PDMS) devices. (**A**) design the device in computer-aided drafting (CAD) software and translate to a negative mold in the software. (**B**) Use photoresist layer (red hatch) on a silicon wafer (solid green) strategically blocked from light with a transparency film (solid gray) (**C**) After light treatment, the reusable negative mold silicon wafer is ready for use. (**D**) Curing a 10:1 ratio of base and curing agent in a vacuum forms the device shape (blue double line hatch). (**E**) The device is cut out of the mold and holes punctured (**F**) PDMS is reversibly attached to a glass slide (solid dark gray) with heat. Figure partially created with BioRender.com.

**Figure 9 ijms-24-10882-f009:**
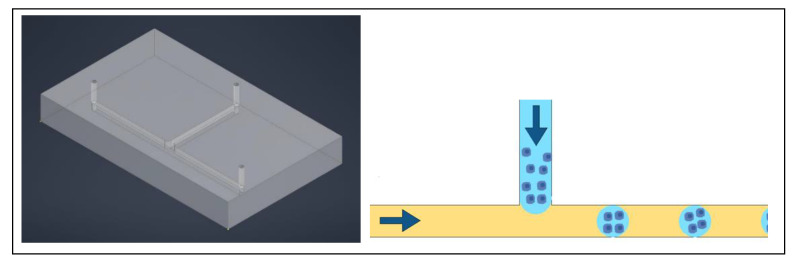
**Left**: simulation of Poly(dimethyl siloxane) (PDMS) chip with T junction, two inlets perpendicular to each other and one outlet. **Right**: oil phase (yellow) flowing from right to left creates droplets of the cell-matrix phase (blue) that flows down and perpendicular to the oil phase.

**Figure 10 ijms-24-10882-f010:**
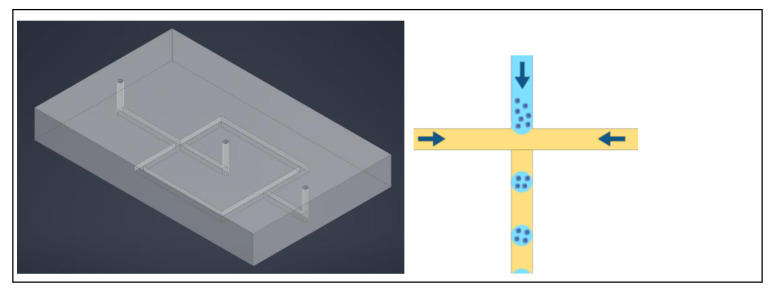
**Left**: simplified simulation of Poly(dimethyl siloxane) PDMS chip with a cross junction, two inlets on the right and one outlet on the left. **Right**: the oil phase (yellow) flows perpendicular to the cell-matrix phase (blue) from opposite sides, creating droplets of the cell-matrix phase.

**Figure 11 ijms-24-10882-f011:**
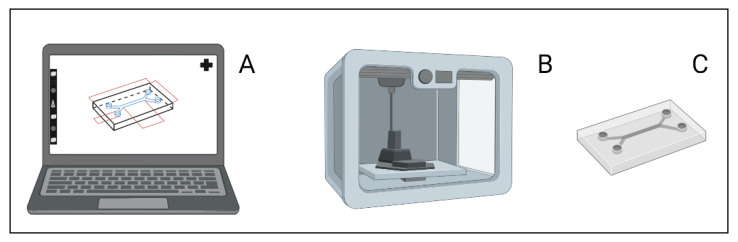
(**A**) A microfluid device is designed using three dimensional (3D) computer aided drafting (CAD) software. (**B**) The design is then printed using powder or resin in a 3D printer. (**C**) The microfluidic chip is ready for use. Figure created with BioRender.com.

**Figure 12 ijms-24-10882-f012:**
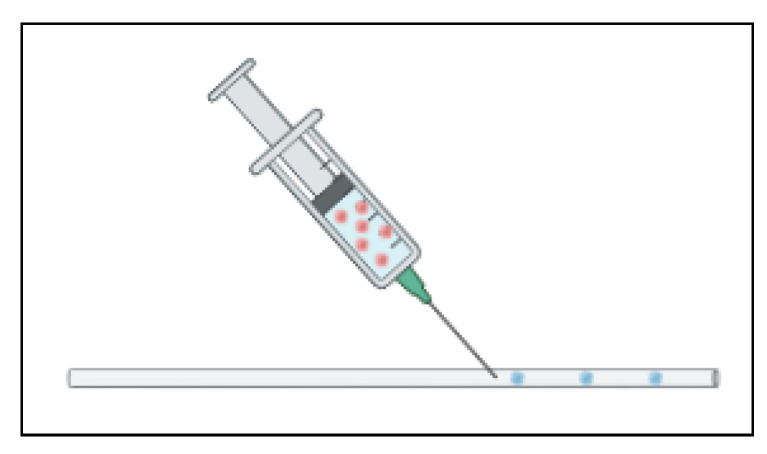
A needle inserted into a tube to inject coated cells utilizes microfluidic fluid flow properties. The cells (red) are in a coating mixture (blue) which then forms the droplets.

**Figure 13 ijms-24-10882-f013:**
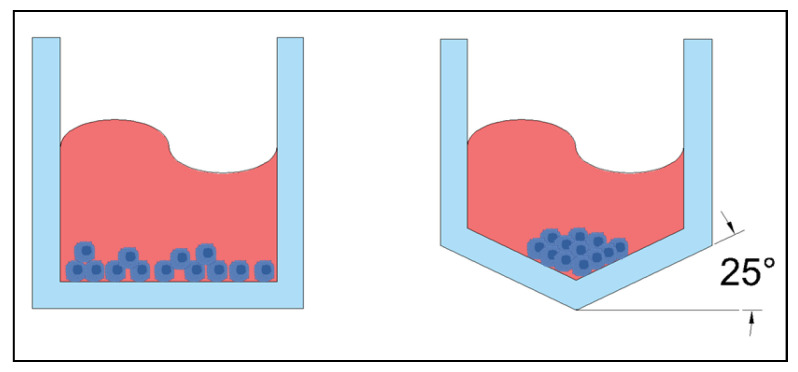
Left: standard flat bottom microwell. Right: concave microwell with the bottom at 25° incline.

**Table 1 ijms-24-10882-t001:** Summary of Device Fabrication Methods.

Method	Primary Advantages	Primary Limitations
Soft lithography (PDMS)	Compatible with imaging Rapid prototyping	Small molecule absorption May complicate downstream analysis
3D printing	Material choice Rapid prototyping	Channels of a specific size may collapse with specific materials
Injection and milling	Material choice Mass production	Not suitable for all geometries without layering
Repurposed lab material	Reduce need for specific equipment and user expertise	Limits complexity of designs Potentially limit reproducibility
Combination of methods	Opportunity to optimize per application Minimize limitations	May require increased expertise

## Data Availability

No new data were created or analyzed in this study. Data sharing is not applicable to this article.

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
