# Peer review of "Microfluidics and Organoids, the Power Couple of Developmental Biology and Oncology Studies"

_ijms, 2023, doi:10.3390/ijms241310882_

Round 1

Reviewer 1 Report

1) There are many facets of working with organoids that are missing. The authors need to review the field properly. 

PMID: 33578886

https://doi.org/10.1038/s41467-020-19058-4

2) Section 2 is not properly organized. It needs to be broken up in terms of the taxonomy of the design.

3) What is the relevance here in terms of the pancreatic cancer? Seems like a general purpose review. 

Reviewer 2 Report

The paper “Microfluidics and organoids, the power couple of developmental biology and oncology studies” by Hetzel et al reviews the microfluidic technology and innovations for organoid investigations. The topic is interesting, but the organization of the article is confusing. Overall, the number of works cited in low in such a broad topic, and the citation is somewhat biased. It is recommended the article should be significantly revised before further consideration.

1. It is recommended that the authors can provide titles of sub-sections in each section, so it might be easier to follow.

2. It is recommended that the authors can first define organoid and distinguish its differences with spheroid, 3D culture, and organ-on-a-chip. It seems like the authors define “organoid” pretty broadly, but different standards have been used in different papers in the field.

3. The quality of the figures is overall not great. It is recommended that the authors can improve the figures. Getting the authorization to use some good original plots from the cited papers might be helpful as well.

4. Overall, the number of cited papers is too low for reviewing such a well-studied field. In addition, many cited works are not representative in the field.

5. Hanging droplet methods for organoid culture should be mentioned. Some relevant papers are listed below:

“ High-throughput 3D spheroid culture and drug testing using a 384 hanging drop array”

“ A microfluidic hanging drop-based spheroid co-culture platform for probing tumor angiogenesis”

“ A simple hanging drop cell culture protocol for generation of 3D spheroids”

“ Reconfigurable microfluidic hanging drop network for multi-tissue interaction and analysis”

6. When the authors talk about the compatibility of 3D organoids with staining, label-free methods to assess their properties have been developed. Some relevant papers are listed below:

“Label-free characterization of organoids with quantitative confocal Raman spectral imaging”

”Direct and Label-Free Cell Status Monitoring of Spheroids and Microcarriers Using Microfluidic Impedance Cytometry”

“Label-free redox imaging of patient-derived organoids using selective plane illumination microscopy”

“Label-free Estimation of Therapeutic Efficacy on 3D Cancer Spheres Using Convolutional Neural Network Image Analysis

“Spectroscopic label-free microscopy of changes in live cell chromatin and biochemical composition in transplantable organoids”

7. Regarding Cancer stem cells (CSCs), it is more accepted that single-cell derived spheroid is a better way to define CSCs. The ref#20 cited by the authors is not a representative work in the field of 3D culture of CSCs. A few representative works in this field are listed:

“High-Throughput Single-Cell Derived Sphere Formation for Cancer Stem-Like Cell Identification and Analysis”

“Enhanced enrichment of prostate cancer stem-like cells with miniaturized 3D culture in liquid core-hydrogel shell microcapsules”

“The enhancement of cancer stem cell properties of MCF-7 cells in 3D collagen scaffolds for modeling of cancer and anti-cancer drugs”

8. The next paragraphs citing Ref#21 and 22 also do not cite representative works in this field. There are many good papers earlier than the cited ones having demonstrated the idea.

9. Not sure why the authors want to include CTCs in this paper.

10. In the fabrication section, the authors mentioned some channel designs (Fig. 9 and 10). Those are off topic and confusing.

11. In Table 1, "PDMS" is not comparable with “3D printing”, and “Injection and milling”. “Soft lithography” might be compared with “3D printing”, and “Injection and milling”. In addition, “Available lab material” is also confusing.

There are typos and grammar issues in the paper. Language editing is needed.

Round 2

Reviewer 2 Report

The authors addressed the comments.

Author Response

Thank you for the feedback.